# Trends in outcomes used to measure the effectiveness of UK-based support interventions and services targeted at adults with experience of domestic and sexual violence and abuse: a scoping review

Sophie Carlisle ,[1] Annie Bunce,[2] Matthew Prina,[3] Elizabeth Cook ,[2] Estela Capelas Barbosa,[4] Sally McManus,[2,5] Gene Feder ,[6] Natalia V Lewis [6]

For numbered affiliations see end of article.

**Correspondence to**
Dr Sophie Carlisle;
sophie.r.carlisle@kcl.ac.uk

## ABSTRACT

**Objectives** In the UK, a range of support services and interventions are available to people who have experienced or perpetrated domestic and sexual violence and abuse (DSVA). However, it is currently not clear which outcomes and outcome measures are used to assess their effectiveness. The objective of this review is to summarise, map and identify trends in outcome measures in evaluations of DSVA services and interventions in the UK.

**Design** Scoping review.

**Data sources** MEDLINE, EMBASE, PsycINFO, Social Policy and Practice, ASSIA, IBSS, Sociological abstracts and SSCI electronic databases were searched from inception until 21 June 2022. Grey literature sources were identified and searched.

**Eligibility** We included randomised controlled trials, non-randomised comparative studies, pre–post studies and service evaluations, with at least one outcome relating to the effectiveness of the support intervention or service for people who have experienced and/or perpetrated DSVA. Outcomes had to be assessed at baseline and at least one more time point, or compared with a comparison group.

**Charting methods** Outcome measures were extracted, iteratively thematically grouped into categories, domains and subdomains, and trends were explored.

**Results** 80 studies reporting 87 DSVA interventions or services were included. A total of 426 outcome measures were extracted, of which 200 were used more than once. The most commonly reported outcome subdomain was DSVA perpetration. Cessation of abuse according to the Severity of Abuse Grid was the most common individual outcome. Analysis of temporal trends showed that the number of studies and outcomes used has increased since the 1990s.

**Conclusions** Our findings highlight inconsistencies between studies in outcome measurement. The increase in the number of studies and variety of measures suggests that as evaluation of DSVA services and interventions matures, there is an increased need for a core of common, reliable metrics to aid comparability.

**Protocol registration** https://osf.io/frh2e.

## STRENGTHS AND LIMITATIONS OF THIS STUDY

⇒ This scoping review adhered to Preferred Reporting Items for Systematic Reviews and Meta-Analyses 2020 guidelines to robustly and systematically map effectiveness outcome measures used in evaluations of support interventions and services for people who have experienced and/or perpetrated domestic and sexual violence and abuse (DSVA).

⇒ It included an extensive grey literature search to be comprehensive and inclusive.

⇒ An advisory group consisting of representatives from six third sector organisations has been involved in the development of the review.

⇒ The review has not included services or interventions for children who have experienced DSVA, interventions aimed at staff training, and is limited to the UK.

⇒ Further research is urgently required to work towards agreement on which outcome measures should be prioritised and adopted, while also avoiding burden on systems and people experiencing DSVA.

## INTRODUCTION

Domestic and sexual violence and abuse (DSVA; comprising a range of different behaviours and experiences, see online supplemental appendix 1) represents a significant international public health issue.[1–3] Approximately one in three women and one in seven men in England and Wales have experienced domestic violence and abuse in their lifetime.[4] Globally, over 35% of women have experienced either physical and/or sexual intimate partner violence or non-partner sexual violence.[5] Poverty, homelessness and insecure immigration status heighten DSVA vulnerability,[6 7] and the risk of experiencing

DSVA is also higher for younger adults, lesbian, gay, bisexual and transgender (LGBT) people,[8–11] and people with long-term illness, disability or mental health problems.[12 13] The UK does not currently routinely collect data on perpetrators; however, studies have found that the prevalence of DSVA perpetration ranges from 11.6% to 80.0% depending on country, gender and definition of DSVA.[14 15] Perpetrators are typically male[16] and are often serial perpetrators with more than one victim.[17 18] The impacts of DSVA are profound and diverse. As well as visible and immediate consequences such as physical injury or death, DSVA has been associated with longer-term physical health[19–25] and mental health problems.[26–29] Other areas impacted by DSVA include homelessness,[30] access to finances, employment and education,[31] and increased demand on the criminal justice system,[32] social services[33] and police.[34]

Given the wide-ranging impacts of DSVA, the support options available to people who have experienced it are diverse. Interventions and services include refuges, outreach, counselling, financial or legal advice, helplines and advocates. In the UK, these are often provided by third sector organisations (organisations that are neither public nor private sector, including voluntary and community organisations; see online supplemental appendix 1) but can also be provided by services located in the public and private sectors. Some services are targeted at particular groups with heightened vulnerability to DSVA, such as LGBT+ and black and minority ethnic (BME) victim-survivors or victim-survivors with disability; however, provision for these groups is stark and disproportionately underfunded.[35–37] Several theoretical frameworks have been developed to explain how such support services and interventions may work. For instance, the socioecological model of violence against women highlights key risk factors at the individual, relationship, community and societal levels, identifying potential intervention points for preventing and responding to violence.[38] One of the risk factors identified by Heise is social isolation, therefore one component of a support intervention may be to increase social support and build relationships. Sullivan's conceptual model of domestic violence support services[39] builds on the Conservation of Resources theory, which suggests that psychological distress following trauma is influenced by the loss of economic, social and interpersonal resources central to well-being.[40–42] Based on this, Sullivan provides an exemplar model for support services which outlines eight common programme activities aimed at creating communities that value their members and promote well-being, through intrapersonal and interpersonal changes, which impact a range of intrapersonal and interpersonal social and emotional well-being outcomes.

Previous systematic reviews of the global evidence have found benefits of such interventions on a range of social and emotional well-being related outcomes, including housing interventions improving mental health, perceived safety and stress,[43] economic interventions reducing levels of domestic violence and increasing empowerment,[44] advocacy interventions improving quality of life and depression[45] and psychological therapies reducing depression and anxiety.[46] Service evaluations and annual reports produced by local DSVA organisations have also found associations between service use and improved perceptions of safety, quality of life, health and well-being, and confidence.[47–51] Finally, reviews of perpetrator programmes (ie, interventions targeted at people who use DSVA) have provided some evidence of reductions in the perpetration and experience of abusive behaviours, although highlighted methodological issues.[52 53]

Despite existing reviews identifying potential benefits of specific types of support services and interventions, there is limited understanding of which outcomes and outcome measures are currently being used both within and across UK-based DSVA support services and interventions when assessing their effectiveness.[54 55] The specific outcome measures that are being used appear to vary, making it difficult to compare and synthesise the overall evidence.[56] Previous research in the UK and US literature that has explored outcome measurement in the DSVA field has noted the diversity of reported outcomes[57 58] and highlighted various issues and difficulties surrounding outcome measurement, which contributes to this diversity.[59 60] These studies point to the differing priorities of funders and service providers, and the diversity of specific DSVA service goals and objectives, which are often multiple and complex, as key drivers behind the range of outcomes being used. This literature also includes discussions on what should be measured, with several pushing for the extension and diversification of outcomes and measurement strategies, potentially contributing to the diversity of outcomes being used. Identifying the most commonly used outcome measures will allow for meta-analysis, leading to more rigorous evaluations of the literature, and may also guide interventions and services on which outcome measures to use. Scoping reviews present an opportunity to map the evidence and are ideal for summarising heterogeneous evidence.[61 62] This scoping review aims to summarise and map trends in the outcome measures being used to evaluate UK-based support interventions and services for DSVA.

## Objectives
Specific objectives of this scoping review include:
1. Identify and thematically group outcome measures that have been used to assess the effectiveness of UK-based support interventions and services for adults who have experienced and/or perpetrated DSVA.
2. Identify which outcome measures have been used most frequently.
3. Explore whether the outcome measures differ by the type of victim-survivor or perpetrator support; DSVA type; source of the study; study setting and over time.

## Review question
What outcomes relating to effectiveness have been measured in intervention studies and service evaluations

of UK-based support programmes for adults who have experienced and/or perpetrated DSVA?

## METHODS

### Protocol and registration

The protocol was developed according to the Preferred Reporting Items for Systematic Reviews and Meta-Analysis Extension for Scoping Reviews (PRISMA-ScR)[63] and published on Open Science Framework: https://osf.io/frh2e.[64] The PRISMA-ScR checklist is available in online supplemental appendix 2.

### Eligibility criteria

We included any UK-based secondary or tertiary prevention support services and interventions aimed at adults who had experienced or perpetrated DSVA. Primary prevention services and interventions were excluded as they aim to prevent DSVA before it occurs, and therefore focus on individuals or populations who have not necessarily experienced violence. Interventions and services that were not specifically aimed at DSVA, such as general trauma interventions, were only included if the majority of participants (>50%) had experienced DSVA. Randomised controlled trials, non-randomised comparative trials, pre–post studies and service evaluations that reported effectiveness outcomes at two or more time points or made comparisons to another group were included, so that cause and effect could be inferred. Perpetrator programmes were considered for several reasons. First, including them allows the measurement of DSVA without placing the burden on people who have experienced DSVA to change someone else's behaviour. This ties into the UK government's new perpetrator strategy,[65] which intends to place the onus on perpetrators to change their behaviour, alongside the recovery of those who have experienced DSVA. Finally, perpetrator programmes provide support for DSVA perpetrators to change their behaviour, and many offer associated support to (ex)partners or referral to appropriate support. Only English language studies were eligible for inclusion. Given the focus on UK-based support services and interventions, we considered the impact of this restriction on the comprehensiveness of the review to be minimal; however, this does mean that the results of this review cannot be generalised to contexts outside of the UK. Details of the full eligibility criteria can be found in table 1.

### Information sources and search strategy

MEDLINE, EMBASE, PsycINFO, Social Policy and Practice, ASSIA, IBSS, Sociological abstracts and SSCI electronic databases were searched from inception until 21 June 2022. Grey literature sources were also identified, including searching electronic grey literature databases (National Grey Literature Collection, EThOS, Social Care Online and the Violence Against Women Network), a call for evidence which was shared with research networks and UK-based DSVA services and organisations and searches

of relevant UK charity and organisation websites. Finally, backward and forward citation tracking of all included studies, as well as reference searching of identified systematic reviews, was undertaken to identify further relevant studies. The search results were exported into EndNote and duplicates were removed. An example of the search strategy for the peer review search and the grey literature search can be found in online supplemental appendix 3.

### Selection of sources of evidence

The de-duplicated citations were uploaded into Rayyan.[66] Titles and abstracts, followed by full-texts, were screened according to the inclusion and exclusion criteria. A 20% sample was independently screened by a second reviewer at each stage, and all disagreements were resolved by discussion, with a third reviewer where needed. Risk of bias assessments is not necessary for scoping reviews.[61]

### Data charting process and data items

Data charting took place using a piloted Excel spreadsheet. Data items extracted included the study citation, study design, participant demographics, description of the service or intervention and comparison (if applicable), setting and the outcome measures used. All outcomes that were used to assess the effectiveness of DSVA support interventions and services were extracted. Data extraction and charting were carried out by one reviewer and checked for errors by a second reviewer. Where data were missing, corresponding authors were contacted and asked to supply said data. After 2 weeks, if there had been no response, corresponding authors were contacted a second time and given a minimum of an additional 2 weeks to respond.

### Synthesis of results

A narrative synthesis was conducted to summarise the included DSVA interventions and services and the effectiveness outcomes they measured. The unit of analysis was the outcome measure(s) reported. DSVA interventions and services were grouped by the type of support offered, based on the intervention and service types outlined in the protocol. Where studies reported more than one form of support as part of the intervention or service, they were classified as 'multi-service'. Where the same intervention or service produced multiple reports (eg, from different years), these were combined into one entry during data charting so that all outcomes ever reported by that service were extracted but were not double counted if multiple reports used the same outcome.

Effectiveness outcomes reported by studies were grouped thematically by type to create categories, domains and subdomains. The groupings and the names for the subdomains, domains and categories were developed iteratively. All outcome measures were first listed in an Excel spreadsheet. This included standardised and unstandardised questionnaires and single-item measures. A column was added, describing what the outcome measured (eg, the Beck Depression Inventory[67] measures

**Table 1**  PICO inclusion and exclusion criteria

| | Inclusion | Exclusion |
|---|---|---|
| Population | ► Adults (16 years or older) who have experienced DSVA<br>► Adults (16 years or older) who have perpetrated DSVA | ► Children<br>► Adults who have not experienced and/or perpetrated DSVA |
| Intervention (and service) | ► Any secondary or tertiary prevention intervention and/or service for DSVA including but not limited to housing (eg, refuges, housing workers, resettlements), Advocacy (eg, Independent Domestic Violence Advisers/Advocates, Independent Sexual Violence Advisers/Advocates), outreach, open access (eg, helplines, drop-ins, online chats), psychological support (eg, support groups, counselling, befriending), legal support, financial support, multi-agency risk assessment conferences and police-based services<br>► Perpetrator programmes (see online supplemental appendix 1)<br>► Entry to the intervention had to be determined by either the experience of DSVA (for victim-survivor support interventions) or perpetrating DSVA (for perpetrator programmes) | ► Pharmacological<br>► Primary prevention<br>► Not primarily aimed at people with experience of DSVA |
| Comparison | ► RCTs:<br>  – Another type of included intervention or service<br>  – Usual care<br>  – No intervention or service<br>► Pre–post designs:<br>  – Before and after the intervention or service | ► No comparison |
| Outcome | ► Any outcomes used to measure effectiveness of DSVA support services and interventions | ► Those not focused on the effectiveness of the intervention or service (eg, process evaluation outcomes such as satisfaction, staff training, etc) |
| Setting | ► UK based | ► Not UK based |
| Study design | ► RCT<br>► Non-randomised comparative<br>► Pre–post<br>► Service evaluation | ► Cross-sectional, case–control, case study<br>► Qualitative<br>► Letters to the editor, think pieces, descriptive only |
| Dates | ► 1982–present | ► Pre-1982 |

DSVA, domestic and sexual violence and abuse; RCT, randomised controlled trial.

depression). These sub-domains were then grouped into broader domains (eg, depression is a mental health-related outcome). Where applicable, these were grouped into again even broader categories (eg, mental health is a form of overall health).

A list of all relevant outcomes and domains, and how frequently they were reported was made. Comparisons were made between outcomes reported by studies in different sectors, over time and by service user (ie, perpetrator or victim-survivor) and support type (ie, psychological support, housing, combinations) or violence type (ie, domestic violence and abuse (DVA), sexual violence and abuse (SVA) and childhood sexual abuse (CSA)).

**Patient and public involvement**

An advisory group was set up comprising representatives from six specialist DSVA organisations who are involved in the delivery, planning, funding or support of specialist DSVA support services in the UK. Specifically, the group included representatives from two second-tier (ie, organisations that support front-line services but do not provide services themselves) domestic abuse organisations, one second-tier organisation for violence against BME women and girls, one domestic abuse organisation that provides a range of front-line services, one service focusing specifically on supporting male victims, working with perpetrators of domestic violence and working with young people using violence in close relationships and one service focusing specifically on sexual violence and abuse. The group inputted to the design of the study

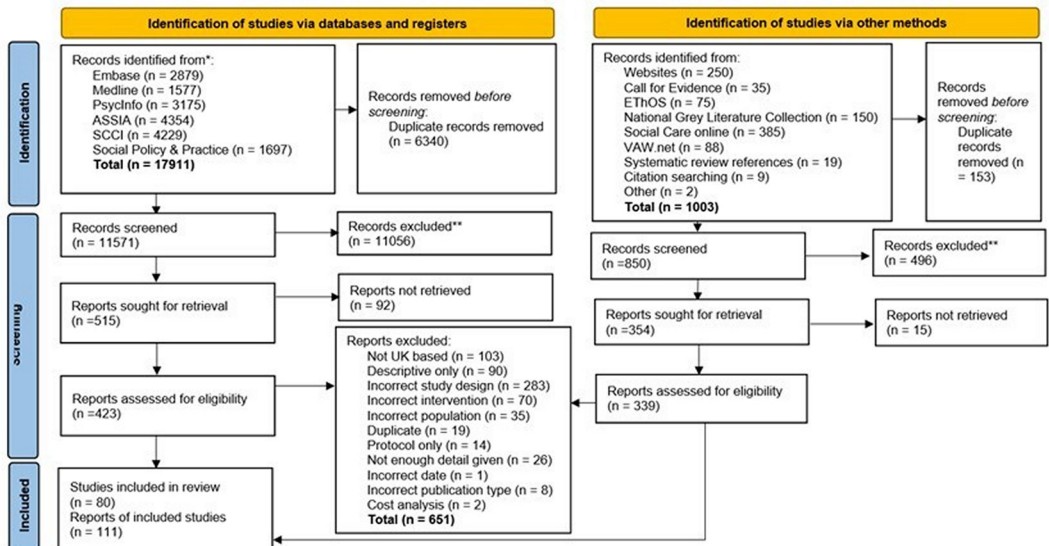

**Figure 1** Preferred Reporting Items for Systematic Reviews and Meta-Analyses flow diagram.

protocol including refinement of the review question and review aims, based on their experiences of service delivery and priorities in terms of addressing knowledge gaps. The group also provided valuable contextual insight regarding the challenges in measuring the effectiveness of support services in the third sector. Their input resulted in several changes, including the broadening of the scope of this review to all effectiveness outcomes, rather than a narrower focus on only outcomes relating to reducing violence, to reflect the priorities of the sector. Provisional findings of the review were shared with the group, and their thoughts regarding the interpretation of the results were discussed, which fed into the manuscript write-up. The finalised review will be disseminated to the advisory group.

## RESULTS

### Selection of sources of evidence

After duplications were removed, a total of 12 431 records and 869 reports were screened for inclusion. 80 studies from 111 reports were included in this review (see figure 1). Data extracted from the included studies are summarised in online supplemental appendix 4 and figure 2.[51 68–177]

### Characteristics of sources of evidence

The 80 studies described a total of 87 DSVA interventions or services, with seven studies describing both a perpetrator programme and an associated support service for (ex)partners of participating perpetrators. All studies were conducted between 1991 and 2022.

### Description of studies

#### Study design

Many of the included studies were mixed-method service evaluations; for these studies, only the methodology relevant to the included outcomes is reported as the other methods do not meet the review eligibility criteria (eg, qualitative, case studies). The most common study design was pre–post single group designs (n=53), of these 14 included additional time points at mid-intervention or follow-up. Eight studies were RCTs, and 10 used a non-randomised comparative design. For one study, the design could not be determined. Eight studies used secondary analysis of datasets, which collated data from multiple services. These were primarily publications produced by SafeLives, a UK-based charity dedicated to ending DSVA, using their Insights system, an outcome measurement tool which allows services across England and Wales to collect data according to various measurement indicators.

### Perpetrator programmes

Perpetrator programmes were grouped together. They were typically described as behaviour change programmes based on a cognitive behavioural model, and while many included additional elements such as psychological support, these were often described in insufficient detail. Of the 80 included studies, 27 described a perpetrator programme.

### Victim-survivor support interventions and services

Interventions and services providing support for those who had experienced DSVA were grouped by type of support provided, based on initial scoping of the literature. The most common were psychological support (n=21), advocacy (n=14) and multi-service (n=11). Three studies each reported on multi-agency risk assessment conferences (MARACs) and specialist domestic violence police teams. Two studies reported housing interventions or services. One study each reported on sexual violence services, a specialist domestic violence court, helpline services, outreach services and adult health-based services. One study described a women's service linked to an associated perpetrator programme, but the specific type of

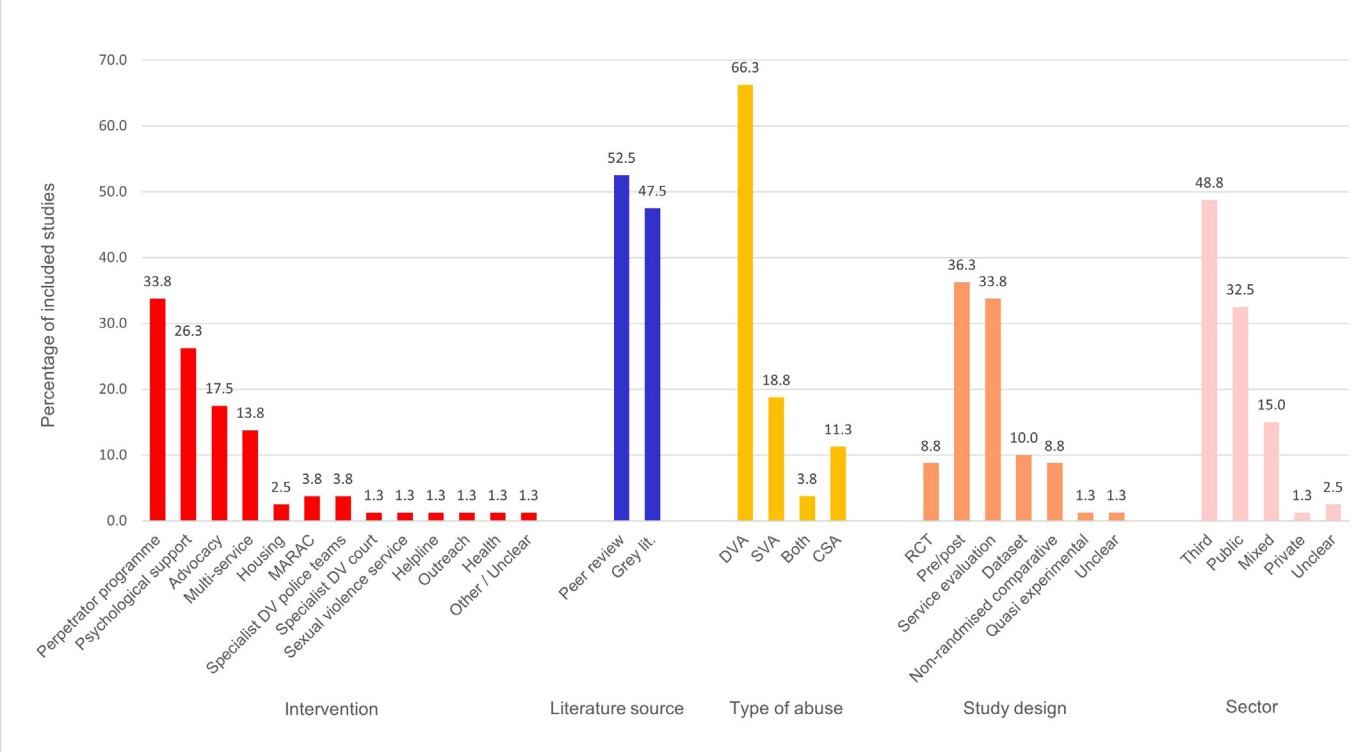

**Figure 2** Bar chart to show the percentage of included studies with each categorisation type. CSA, childhood sexual abuse; DV, domestic violence; DVA, domestic violence and abuse; MARAC, multi-agency risk assessment conference; RCT, randomised controlled trial; SVA, sexual violence and abuse.

support offered was unclear. Several studies evaluated the effectiveness of more than one intervention or service. Of the 60 victim-survivor support interventions and services, two were specifically targeted at male victim-survivors, one at BME women, one at victims of trafficking and one at women with disabilities. There was no evidence for interventions or services targeted specifically at LGBT+ victim-survivors.

### Type of DSVA

The studies were categorised as being primarily aimed at DVA, SVA, both DVA and SVA (DSVA) or CSA. Categorisations were based on how the study or service described itself; however, it should be noted that the type of violence/abuse was often not clearly defined. Thus, these categorisations should be considered as having ambiguous boundaries and interpreted with caution.

The majority (n=53; 66.3%) of the studies were aimed at DVA, while 24 (29.3%) were aimed specifically at SVA. Of these, nine studies were specifically targeted at adults who had experienced CSA. Three studies (3.8%) were aimed at DSVA.

### Source of the studies

Of the 80 studies and 111 reports, 41 studies from 47 reports were located in the peer-reviewed literature, and 39 studies from 64 reports were located in the grey literature.

### Setting

Just under half of the studies were conducted in the third sector (n=39; 47.6%). 26 studies were conducted in the public sector, and 12 spanned multiple sectors. One was conducted in a privately owned prison, and for one study the setting was unclear.

### Types of outcomes

The 80 included studies measured a total of 426 outcomes. Of these, 200 (47.0%) were used more than once. In total, there were 282 unique outcomes, 57 of which were used more than once, and 226 were used only once. The number of effectiveness outcomes measured by each individual study ranged from 1 to 19, with a mean average of 5.23.

Outcomes were grouped thematically into 11 domains and 49 subdomains. The 11 domains included mental health, physical health, general well-being (all of which fall under the broad category of 'health'), the experience of DSVA and the perpetration of DSVA (which fall under the broad category of 'DSVA'), empowerment, socioeconomic circumstances, behavioural outcomes, the victim–perpetrator relationship, parenting and changing perpetrator beliefs and skills (online supplemental appendix 5).

Many of the outcome measures reported spanned two or more domains (figure 3). For instance, 'health' as an overarching category included three domains: mental health, physical health and general well-being. 'Quality

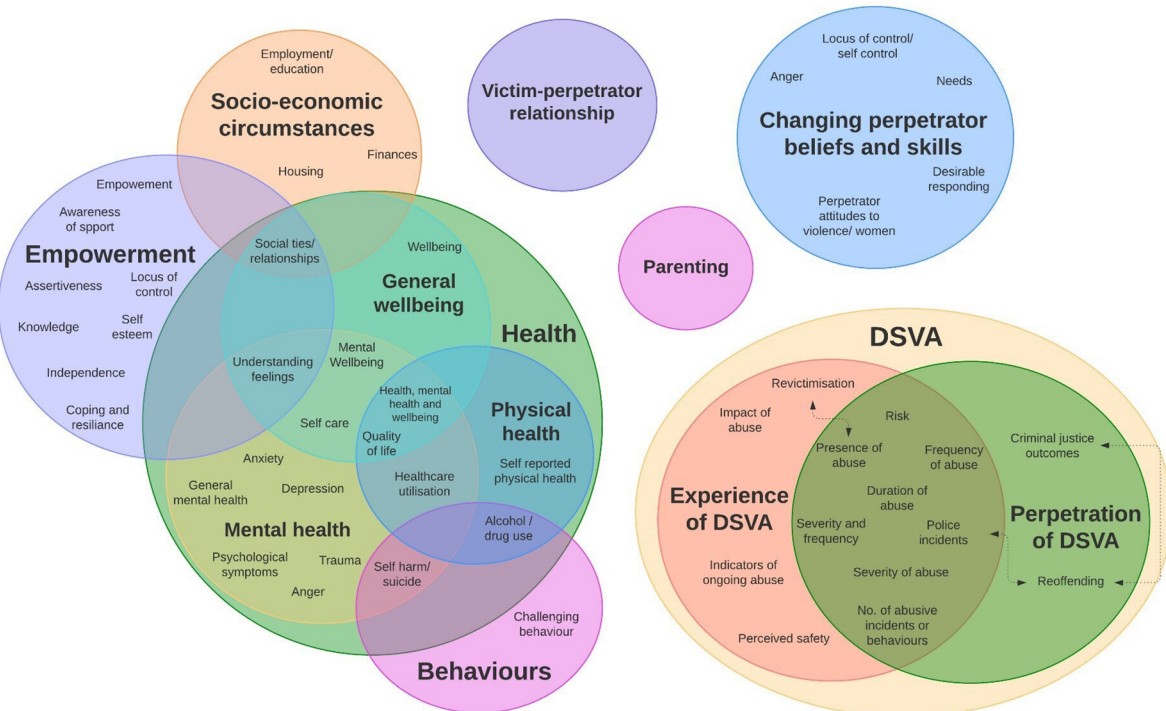

**Figure 3** Venn diagram of outcome categories, domains and subdomains. DSVA, domestic and sexual violence and abuse.

of life' as a subdomain spanned physical health, mental health and general well-being domains, based on the identified outcome measures and their specific items. For example, the 12-item Short Form is a measure of quality of life and general health, which includes items relating to physical health, mental health and general well-being. Similarly, the overarching category of DSVA contained two domains: the experience of DSVA and the perpetration of DSVA. However, many outcome measures could fall into either domain, depending on who the respondent was (ie, the perpetrator or the victim-survivor). For example, the outcome 'presence of abuse' would fall into the 'experience of DSVA' domain if the respondent was the victim-survivor, and the 'perpetration of DSVA' domain if the respondent was the perpetrator. Thus, there was substantial overlap between these two subdomains. Many of the outcomes within the DSVA category, while defined slightly differently, were not distinct. For instance, there is significant overlap between the concept of 'revictimisation' and the outcome 'presence of abuse' when measured using a pre–post design (if the victim-survivor indicates that they are still experiencing abuse postintervention, then this indicates 'revictimisation'). There is similar overlap between reoffending and revictimisation (although we cannot presume a direct overlap, as a perpetrator may have multiple victims) and between 'reoffending' and 'police incidents' (although not all reoffending will be recorded by police therefore this subdomain may be broader).

For each of the 11 domains, the total number of outcomes, the total number of unique outcomes and the number of studies reporting at least one outcome

that fell within that domain were explored. The domain 'perpetration of abuse' has the highest number of studies reporting at least one outcome, the most total outcome measures and the most unique outcome measures (online supplemental appendix 6).

Of the three most common outcome measures across intervention types, all fell within the DSVA category (experience of DSVA/perpetration of DSVA; table 2). Cessation of abuse according to the Severity of Abuse Grid (SAG)[178] was the most common outcome (n=11), followed by the presence of abuse (n=10) and the severity of abuse (n=8), both also measured by the SAG. The Rosenberg Self-Esteem Scale[179] and two more measures of DSVA from the SAG (multiple types of abuse; abuse that is escalating in severity or frequency) were each reported by seven studies.

### Patterns in the use of outcome

The table in online supplemental appendix 6 summarises the patterns in outcome domains according to perpetrator programmes, victim support type, DSVA type, source of literature and setting/sector.

### Outcomes in perpetrator programmes

Of the 27 perpetrator programmes, 14 reported at least one outcome relating to changing perpetrator behaviour, and 21 reported at least one relating to the perpetration of DSVA.

### Outcomes by victim support type

The most common outcome domains for psychological support interventions and services were mental health

**Table 2** Most commonly reported outcome measures

| Outcome | Subdomain | Domain | Studies (n) |
|---|---|---|---|
| **Overall** | | | |
| 1. Cessation of abuse measured by the SAG[178] | Presence of abuse | Experience of DSVA/perpetration of DSVA | 11 |
| 2. Presence of abuse measured by the SAG*[178] | Presence of abuse | Experience of DSVA/perpetration of DSVA | 10 |
| 3. Severity of abuse measured by the SAG*[178] | Severity of abuse | Experience of DSVA/perpetration of DSVA | 8 |
| 4. RSES[179] | Self-esteem | Empowerment | 7 |
| 5. Multiple types of abuse measured by the SAG[178] | Presence of abuse | Experience of DSVA/perpetration of DSVA | 7 |
| 6. Abuse that is escalating in severity or frequency as measured by the SAG[178] | Severity and frequency of abuse | Experience of DSVA/perpetration of DSVA | 7 |
| **Peer review versus grey literature** | | | |
| Grey literature | | | |
| 1. Cessation of abuse measured by the SAG[178] | Presence of abuse | Experience of DSVA/perpetration of DSVA | 9 |
| 2. Presence of abuse measured by the SAG*[178] | Presence of abuse | Experience of DSVA/perpetration of DSVA | 9 |
| 3. Severity of abuse measured by the SAG*[178] | Severity of abuse | Experience of DSVA/perpetration of DSVA | 8 |
| Peer review | | | |
| 1. RSES[179] | Self-esteem | Empowerment | 6 |
| 2. BDI[67] | Depression | Mental health | 5 |
| 3. CORE-OM[193] | Psychological issues | Mental health | 5 |
| **Sector** | | | |
| Third | | | |
| 1. Cessation of abuse measured by the SAG[178] | Presence of abuse | Experience of DSVA/perpetration of DSVA | 9 |
| 2. Presence of abuse measured by the SAG*[178] | Presence of abuse | Experience of DSVA/perpetration of DSVA | 9 |
| 3. Severity of abuse measured by the SAG*[178] | Severity of abuse | Experience of DSVA/perpetration of DSVA | 8 |
| Public | | | |
| 1. BDI[67] | Depression | Mental health | 5 |
| 2. RSES[179] | Self-esteem | Empowerment | 4 |
| 3. SCL-90R[194] | Psychological issues | Mental health | 4 |
| Mixed | | | |
| 1. Cases discontinued | Criminal justice outcomes | Perpetration of DSVA | 2 |
| 2. PHQ[195] | Depression | Mental health | 2 |
| 3. GAD-7[196] | Anxiety | Mental health | 2 |
| 4. Cambridge Crime Harm Index[197] | Severity of abuse | Experience of DSVA/perpetration of DSVA | 2 |
| 5. PC-PTSD[198] | Trauma | Mental health | 2 |
| No further post-MARAC police complaints or call-outs | Police incidents | Experience of DSVA/perpetration of DSVA | 2 |
| No further post-MARAC police complaints | Police incidents | Experience of DSVA/perpetration of DSVA | 2 |
| No further post-MARAC police call-outs | Police incidents | Experience of DSVA/perpetration of DSVA | 2 |

*Reported separately for physical abuse, sexual abuse, harassment and stalking, and jealous and controlling behaviour.
BDI, Beck Depression Inventory; CORE-OM, Clinical Outcomes in Routine Evaluation Outcome Measure; DSVA, domestic and sexual violence and abuse; GAD-7, Generalised Anxiety Disorder Questionnaire; MARAC, multi-agency risk assessment conference; PC-PTSD-5, Primary Care PTSD Screen; PHQ, Patient Health Questionnaire; RSES, Rosenberg Self-Esteem Scale; SAG, Severity of Abuse Grid; SCL-90R, Symptom Checklist 90 revised.

(n=17) and empowerment (n=10), while advocacy, multi-service interventions and studies of MARACs primarily reported on experience of DSVA (n=12, n=8 and n=3, respectively). Specialist DV police teams reported equally on the experience of and the perpetration of DSVA (n=2 each).

### Outcomes by type of DSVA
Of the studies primarily focused on DSVA, the majority reported at least one outcome relating to the experience of DSVA (n=40; 75.5%) and the perpetration of DSVA (n=41; 77.4%). Just under half of studies of adult SVA (n=7; 46.7%), and all studies of CSA, reported mental health outcomes (n=9; 100%).

### Outcomes by source of studies
While the split of included studies from the grey literature and peer-reviewed literature was relatively even, studies located within the grey literature were more diverse. In total, grey literature studies reported 254 outcome measures, 178 of which were unique. In comparison, peer-reviewed studies reported a total of 172 outcomes, 125 of which were unique. Peer-reviewed studies had more of a tendency to report at least one outcome relating to mental health (n=22; 53.7%), while grey literature studies were more likely to report outcomes relating to the experience of DSVA (n=29; 74.4%).

### Outcomes by setting
Two-thirds of studies set in the third sector reported outcomes relating to the experience of DSVA subdomain (n=27; 69.2%). Perpetration of DSVA was the most common outcome subdomain for interventions and services with a mixed setting (n=11; 91.7%), while mental health was the most common subdomain for those in the public sector (n=13; 50%).

### Outcomes 1991–2022
Evaluations of DSVA support services and interventions have become more frequent over time. Nearly half (48.8%) of included studies were published in 2015–2022, with over a quarter (26.3%) published in 2019–2022 (online supplemental appendix 7).

### DISCUSSION
This study used scoping review methodology to map which outcomes have been used in studies assessing the effectiveness of a range of support interventions and services for people who have experienced or perpetrated DSVA. A total of 426 outcome measures were used in 80 papers and reports, describing 87 DSVA interventions and services. Less than half of the measures were used more than once.

The most common outcome domain was the perpetration of DSVA, followed closely by the experience of DSVA, both of which included outcomes relating directly to abusive and violent behaviours. This demonstrates that the measurement of DSVA has been prioritised

when assessing service and intervention effectiveness, and is consistent with the common goal of many DSVA support services and organisations of ending violence and abuse.[180 181] However, 201 and 189 unique outcome measures were charted within these two domains, respectively, showing the lack of consensus regarding the specific outcomes that should be measured. The SAG, which measures several different aspects of abusive behaviours (including the presence of four types of abuse, frequency, severity and cessation of abuse), was the most frequently used outcome measure. It was primarily used by studies drawing on the SafeLives' Insights datasets, although it was employed by several additional studies too. Despite being the most frequently reported outcome, the SAG was still used by only 13 of the 80 studies, illustrating the inconsistencies and variation in measures being used by these evaluations and reports.

Of the 282 unique outcomes reported, 226 were used only once. Although many of these outcomes were similar, they often differed slightly according to the unit or method of measurement, meaning that combining them would be methodologically inappropriate. This further demonstrates the disparate and highly variable nature of outcome measures currently in use. While this may be partly explained by the fact that different sectors and types of support interventions have different priorities, such disparity is evident even *within* sectors and intervention types. For instance, of the 153 different outcome measures reported by third sector studies, only 24.8% were used more than once, with a similarly low proportion in mixed sector and public sector studies (both 30.7%). This indicates a lack of comparability surrounding the best way to assess the effectiveness of support services and interventions and makes synthesis such as meta-analysis less feasible.

Mapping revealed differences between the types of outcome domains frequently reported in the peer-reviewed and grey literature, with peer-reviewed studies focusing more on mental health-related outcomes, and grey literature studies focusing primarily on DSVA. This difference may be explained by the overlap between grey literature and third sector-based studies, and the fact that third sector organisations are heavily tied to the outcomes that funders and commissioners require them to measure. With the focus on wanting to establish and reduce the prevalence of DSVA itself, and the associated focus on the perpetration of violence, it might be that more funders and commissioners are requiring these to be measured or are prioritising these measurements above the wider impacts of DSVA.

Mapping of trends over time showed that as well as a general increase in the number of studies being conducted, there appeared to be increased diversity in outcomes measured. The two most common outcome categories, DSVA and health, have been measured fairly consistently over the past 30 years, while domains such as empowerment, the victim-perpetrator relationship and parenting have become more common in the past

decade. This is likely reflective of an increased knowledge of the wide-ranging impacts of DSVA and its ripple effects, increased recognition of the need for holistic support and the move towards survivor-centred outcome measurement and consultation with those with lived experience.[182]

Overall, this review has demonstrated that the types of outcomes measured when assessing the effectiveness of support interventions and services for people who have experienced DSVA vary widely. There are several potential reasons for this diversity in outcome measures. First, as previously mentioned, DSVA has many impacts, including on physical health, mental health, housing, finances, relationships and many more. Therefore, there are many avenues through which support services and interventions could have impact and improve outcomes. Indeed, in their review Sprague et al[57] noted that many DSVA services have multiple goals that cannot be assessed by a single outcome, thus multiple outcomes are required. Second, differences between the priorities of funders, services and service users, and researchers in terms of outcome measurement likely result in diversity in outcomes and differences in outcome measures used between sectors.[58 59] Third, there is currently a lack of consensus across the field regarding what outcomes should be measured and, importantly, how. For instance, while it may be agreed that 'safety' is an important outcome to assess, how this is operationalised and measured is not consistent.

## Challenges in outcome measurement

There are two linked challenges facing UK-based DSVA services, especially those located in the third sector. First, funding of DSVA services is precarious and diminishing.[183–186] Being awarded funding requires evidence of effectiveness. However, the second challenge is that carrying out service evaluations to provide evidence of effectiveness is costly, making it even more difficult for services to secure funding. This is reflected in take-up of the aforementioned SafeLives' Insights system, which captures rich data and has likely contributed to more consistent reporting of outcomes across England and Wales. However, individual services have to pay to use the system, and the number of services contributing to the dataset each year has reduced, likely due to reduced funding (consultation with stakeholder advisory group).

## Limitations

There are limitations of this review associated with the scoping review methodology. For instance, scoping reviews prioritise breadth over depth. The primary objectives of this review were to map and chart outcome measures being used (ie, the breadth of outcome literature), and so we have not made any assessments regarding the validity or relevance of any particular measure. Therefore, while we have identified the most commonly used outcome measures, that is not to say that these are the most helpful, informative or important to people with lived experience or to front-line service providers.

Further, in line with the scoping review methodology, we have not completed risk of bias assessments for any of the included studies, therefore we cannot comment on the quality of these studies. A further limitation of this review is the unavoidable potential for double counting. For instance, the services that are included in SafeLives' Insights datasets are anonymous. Thus, it is not possible to determine if any services included in the Insights datasets have also published an independent evaluation. Another limitation of this review is that despite a comprehensive search strategy, no evaluations of support services run by by-and-for organisations were identified (see online supplemental appendix 1). This is illustrative of a systemic, cyclical and enduring issue facing these smaller services in particular, which are disproportionally underfunded and lack the resources needed to conduct and publish evaluations, meaning they are under-represented in the literature.[36 187–189] Further, despite the comprehensive search strategy which was designed in collaboration with the advisory group to ensure key search terms were included, the search was not exhaustive, and therefore it is possible that had we selected different databases or included additional terms, we may have identified additional relevant papers. Additionally, because of time and resource restrictions, only 20% of records were dual-screened during the title/abstract and full-text screening stages, which does increase the risk of potential bias in the selection of studies and the risk of relevant studies being incorrectly excluded. Finally, we excluded process evaluation outcomes. Given that there may be causal links between process measures and distal outcomes, it is possible that some relevant outcomes were missed.

## Implications and recommendations

The findings from this scoping review regarding outcomes most commonly measured to assess the effectiveness of DSVA interventions highlight how support services are working to promote survivors' safety and well-being. However, the diversity of the specific outcomes and measurement tools used has implications for researchers, service providers, policymakers and funders.

### Implications and recommendations for researchers

Increased consensus between researchers, service providers, policymakers and funders is needed. This will allow for more meaningful syntheses of the literature, as well as building a larger evidence base, so that a better understanding of the most effective means of support for people who have experienced DSVA can be reached. Thus, further research is needed to extend ongoing work on the development of shared outcome frameworks, to determine which outcomes and outcome measures are most appropriate, valid and relevant, in order to work towards a consensus and build a shared evidence base to enable future meta-analyses. These outcome frameworks should be underpinned by programme theories. We hope our findings will inspire further conversation and exploration of what to measure, when and with whom, and

provide guidance to researchers, service providers and funders striving to strike the balance between reaching a shared consensus and selecting outcomes according to important contextual factors such as the (often multiple and complex) goals of the intervention, the timing of evaluation and the study population, for example. This work should be carried out as a collaboration between people who have experienced abuse and their families, service providers and, in the UK, commissioners, to ensure researchers and evaluators are measuring the outcomes that matter most to the people who matter most. Such approaches have previously been used to develop outcome frameworks in related areas.[56] This work should also consider whether the outcomes and outcome measures most appropriate and relevant vary by different characteristics or populations, such as male victim-survivors, BME populations or those from the LGBT+ community. Any outcome frameworks that are developed should consider this and be adaptable to the different target populations of various services.

### Implications and recommendations for service providers

While we recognise that service providers working in the DSVA field are often overburdened and under-resourced, and doing the best they can with what they have, we would encourage service providers to use the findings from this review to consider their current practices for outcome measurement and the how's and why's underlying them. If resource allows, organisations could assess whether the outcome measurement tools currently being used are fit for purpose and reflect recent shifts in definitions of and responses to domestic abuse, and explore opportunities for improvement and innovation. In the absence of additional resource, service providers should continue to record and monitor the various service-level and individual-level outcomes they are required to measure as accurately and consistently as possible while delivering front-line services.

### Implications and recommendations for policymakers

Ultimately, the extent to which service providers can apply research findings to practice and use them to drive improvements to services depends heavily on policymakers' decisions relating to the provision and funding of services, and monitoring and evaluation. Currently, patchwork and piecemeal funding[36 190] and inconsistent commissioning frameworks contribute to the diversity of outcomes measured and underpin a competitive funding landscape that is not conducive to achieving consensus across the sector. Service providers working to end DSVA have been campaigning for greater consistency in the commissioning of services for many years, and their voices should be used by policymakers to enact meaningful change. Based on discussions with our advisory group throughout the review process, we would suggest that policymakers focus on ensuring any core outcomes framework is implementable on a practical level. Service providers in low-resource settings likely have less scope for

this than others, and it is crucial that any core outcome measurement tools are not too time-consuming for service providers to use and complete. Additionally, policymakers and commissioners should look beyond blunt, immediately measurable outcomes (eg, immediate safety, prosecution and conviction rates) and recognise the value of other outcomes (eg, social and emotional well-being), which may take longer to measure but be a more accurate indication of real change. A suggested avenue for working towards this would be to run a national survivor consultation to draw out what is the most meaningful and/or has made the most difference for survivors. Key findings from this could then be mapped onto the outcomes prioritised by policymakers, service providers and funders, to identify where there are overlaps and where there are disparities that need addressing.

## CONCLUSIONS

A broad range of outcome measures have been used to assess the effectiveness of support interventions for people who have experienced or perpetrated DSVA. There is growing recognition across various sectors of the potential benefits of a core set of outcomes, which include increasing consistency and consensus.[191 192] We need to take stock of the current state of the evidence and reconceptualise what a core set of outcomes should look like across and within sectors. This core outcome set needs to be coproduced by a range of stakeholders, including those with experience of DSVA and those providing front-line support. For this to be feasible, a trade-off needs to occur. Burden needs to be minimised by not requiring excessive data collection, but instead, collection of only the most relevant outcome measures according to the specific purpose of the intervention or service. At the same time, within-service consistency needs to be increased, as well as supporting comparability across sectors. Further research is needed to better understand which outcomes should be prioritised in order to strike this balance.

**Author affiliations**
[1]Health Service and Population Research, King's College London, London, UK
[2]Violence and Society Centre, City University of London, London, UK
[3]Population Health Sciences Institute, Newcastle University, Newcastle upon Tyne, UK
[4]Population Health Sciences, Bristol Medical School, University of Bristol, Bristol, UK
[5]National Centre for Social Research, London, UK
[6]Centre for Academic Primary Care, Population Health Sciences, Bristol Medical School, University of Bristol, Bristol, UK

**Acknowledgements** We thank the representatives from SafeLives, Women's Aid Federation England, Respect, Rape Crisis England and Wales, Refuge and Imkaan who made up our patient and public advisory group, for their contribution to this project.

**Contributors** All authors (SC, AB, MP, EC, ECB, SM, GF and NVL) contributed to the conceptualisation and design of the review. SC carried out searches, screening, extraction, data charting and analysis. AB was the second reviewer and contributed to the screening and extraction stages. SC produced the first draft of the manuscript and subsequent revised versions following valuable input and refinement from the coauthors (AB, MP, EC, ECB, SM, GF and NVL). All authors approved the final version. SC acted as guarantor.

**Funding** This research was supported by the UK Prevention Research Partnership (Violence, Health and Society; MR-VO49879/1), which is funded by the British Heart Foundation, Chief Scientist Office of the Scottish Government Health and Social Care Directorates, Engineering and Physical Sciences Research Council, Economic and Social Research Council, Health and Social Care Research and Development Division (Welsh Government), Medical Research Council, National Institute for Health and Care Research, Natural Environment Research Council, Public Health Agency (Northern Ireland), The Health Foundation and Wellcome. The views expressed in this Article are those of the authors and not necessarily those of the UK Prevention Research Partnership or any other funder. The funder had no input in the design of the protocol or conduct of the review.

**Competing interests** None declared.

**Patient and public involvement** Patients and/or the public were involved in the design, or conduct, or reporting, or dissemination plans of this research. Refer to the Methods section for further details.

**Patient consent for publication** Not applicable.

**Provenance and peer review** Not commissioned; externally peer reviewed.

**Data availability statement** All data relevant to the study are included in the article or uploaded as supplementary information.

**ORCID iDs**
Sophie Carlisle http://orcid.org/0000-0002-9784-9910
Elizabeth Cook http://orcid.org/0000-0002-7608-8702
Gene Feder http://orcid.org/0000-0002-7890-3926
Natalia V Lewis http://orcid.org/0000-0002-4839-6548

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
