## [Reviewer comments · BMJ Open]

ARTICLE DETAILS

TITLE (PROVISIONAL)	Trends in outcomes used to measure the effectiveness of UK-based support interventions and services targeted at adults with experience of domestic and sexual violence and abuse: A scoping review.
AUTHORS	Carlisle, Sophie; Bunce, Annie; Prina, Matthew; Cook, Elizabeth; Barbosa, Estela; McManus, Sally; Feder, Gene; Lewis, Natalia

VERSION 1 – REVIEW

REVIEWER	Benjamin Hine University of West London
REVIEW RETURNED	21-Jun-2023

GENERAL COMMENTS	This manuscript details a meta-analysis exploring the outcome measures utilised by programmes supporting victims and perpetrators of DVSA. It is a robust study, with few flaws. However, I would recommend addressing the following before publication: Introduction - The authors note the prevalence rates for women versus men, but then as the review progresses there is little acknowledgement of how the study findings they reference reflect gendered experiences (or any other demographic variable) for that matter. For example, in the last paragraph on page 6 through 7, they note issues with existing provision and it's evaluation, without noting that this is much worse for services catering for specific groups (i.e., men or LGBT+ survivors). A paragraph on this nuance or a greater threading of this narrative throughout the introduction would be very useful. Method - RE Patient and public involvement – can we have more information as to what type of organisations were included? And who exactly they catered for? This is partially related to my comment above on the introduction Results - Error on page 15 line 27 Discussion - Similarly to the introduction, I wonder if there is scope here, even if just in the limitations/future research directions section, to mention how outcomes might vary/need to vary based on the characteristics of the individual. For example, some measures of perpetration may be more/less relevant in LGBT relationships. Would be worth considering, either formally (i.e., if you have the information to look at this in a detailed way) or informally (i.e., as a future direction).
---

REVIEWER	Ameeta Kalokhe Emory University
REVIEW RETURNED	08-Aug-2023

GENERAL COMMENTS	The article by Carlisle et al is a comprehensive scoping review of measures used to assess effectiveness of interventions for addressing domestic and sexual violence in the UK over the past 3 decades. Overall, the review is very well written. The extension of the scoping review to the gray literature and use of an advisory board made up of third sector stakeholders is a key strength. I have a few comments: Article summary/Intro – given the global audience/readership, please define “third sector” organizations, as this is not a commonly used term in many countries. Introduction – very well developed, strong support for research question. Objective 3 – Please check grammar Methods  - Authors should justify why primary prevention interventions were excluded. You would think many of the outcomes measured would be similar/relevant – i.e. DVSA, health, empowerment - Authors should discuss how general trauma intervention studies were handled – were they included/excluded? - What search terms were used and how were they identified? - When data was missing, how long did article authors get to submit their responses? Results  - Authors should elaborate on the type of health outcomes measured - This sentence needs clarification: “Similarly, whilst the overarching category of DSVAs contained two primary domains: the experience of DSVAs and the perpetration of DSVAs, many outcome measures could fall into either domain, depending on who the respondent was (i.e., the perpetrator or the victim-survivor).” - Page 15 has an inserted formatting error: (Appendix 5Error! Reference source not found.) Figure 1 needs to be reviewed and adjusted as the numbers don't flow/add up in the current form. For example, why are the records screened (11571) different from total identified (17911)? Discussion: Overall well developed. I would add to the limitations that only 20% of records were dually screened at each stage.
--

VERSION 1 – AUTHOR RESPONSE

Reviewer comment	Response	Revised text
Reviewer 1		
1. It is a robust study, with few flaws.	Thank you for the positive feedback and helpful suggestions for revisions.	
2. Introduction - The authors note the prevalence rates for women versus men, but then as the review progresses there is little acknowledgement of how the study findings they	Thank you for your comment. This is a really important point – whether some services are available or are evaluated for some groups but not others, and whether they are more	Introduction Poverty, homelessness, and insecure immigration status heighten DSVAs vulnerability [6, 7], and the risk of experiencing

reference reflect gendered experiences (or any other demographic variable) for that matter. For example, in the last paragraph on page 6 through 7, they note issues with existing provision and it's evaluation, without noting that this is much worse for services catering for specific groups (i.e., men or LGBT+ survivors). A paragraph on this nuance or a greater threading of this narrative throughout the introduction would be very useful.	effective for some groups than others. These are such important issues they warrant a full study, given the range of outcomes already being considered it was beyond our scope in this review. However, we have added more information throughout the introduction section on various demographic factors that heighten vulnerability to DSVA, as well as noting that services specifically dedicated to these groups face the biggest challenges when it comes to funding. We have also added into the results section information on the number of included studies that were specifically targeted at these specific groups, including male victim-survivors, LGBT victim-survivors, BME populations and those with disability. We have not gone into the specific outcomes reported in the services/interventions for these groups as this goes beyond the scope of this scoping review, but we have commented on future work related to this that could be done in the discussion section.	DSVA is also higher for younger adults, lesbian, gay, bisexual and transgender (LGBT) people [8-11], and people with long term illness, disability or mental health problems [12, 13]... Some services are targeted at particular groups with heightened vulnerability to DSVA, such as LGBT+ and Black and Minority Ethnic (BME) victim-survivors or victim-survivors with disability, however provision for these groups is stark and disproportionately underfunded [35-37]. Results Of the 60 victim-survivor support interventions and services, two were specifically targeted at male victim-survivors, one at BME women, one at victims of trafficking, and one at women with disabilities. There was no evidence for interventions or services targeted specifically at LGBT+ victim-survivors.
3. Method - RE Patient and public involvement – can we have more information as to what type of organisations were included? And who exactly they catered for? This is partially related to my comment above on the introduction	This is a good suggestion, thank you. We have added in more details into the PPI section regarding the types of organisations and who they support.	Specifically, the group included representatives from two second-tier (i.e., organisations that support front-line services but do not provide services themselves) domestic abuse organisations, one second-tier organisation for violence against Black and Minority Ethnic (BME) women and girls, one domestic abuse organisation that provides a

		range of front-line services, one service focusing specifically on supporting male victims, working with perpetrators of domestic violence, and working with young people using violence in close relationships, and one service focusing specifically on sexual violence and abuse.
4. Results - Error on page 15 line 27	Thank you for highlighting this error, this has been corrected.	
5. Discussion - Similarly to the introduction, I wonder if there is scope here, even if just in the limitations/future research directions section, to mention how outcomes might vary/need to vary based on the characteristics of the individual. For example, some measures of perpetration may be more/less relevant in LGBT relationships. Would be worth considering, either formally (i.e., if you have the information to look at this in a detailed way) or informally (i.e., as a future direction).	This is a valid point, and we have added a sentence in the implications and recommendations section to say that future work should also consider whether the outcomes that matter most to victim-survivors and to perpetrators varies according to their characteristics and circumstances.	This work should also consider whether the outcomes and outcome measures most appropriate and relevant vary by different characteristics or populations, such as male victim-survivors, BME populations, or those from the LGBT+ community. Any outcome frameworks that are developed should consider this and be adaptable to the different target populations of various services.
Reviewer 2		
6. Overall, the review is very well written. The extension of the scoping review to the gray literature and use of an advisory board made up of third sector stakeholders is a key strength.	Thank you for the positive feedback and constructive suggestions for improvement.	
7. Article summary/Intro – given the global audience/readership, please define “third sector” organizations, as this is not a commonly used term in many countries.	Thank you for this comment. The term “third sector organisation” is included in our glossary in appendix one, which is referred to in the introduction, however we have added a brief in-text definition for clarity.	Organisations that are neither public nor private sector, including voluntary and community organisations
8. Introduction – very well developed, strong support for	Thank you for the positive feedback.	

research question.		
9. Objective 3 – Please check grammar	We have edited objective 3 to improve the grammar and readability.	Explore whether the outcome measures used differ by: the type of victim-survivor or perpetrator support; DSVA type; source of the studies; study setting; and over time.
10. Methods - Authors should justify why primary prevention interventions were excluded. You would think many of the outcomes measured would be similar/relevant – i.e. DVSA, health, empowerment	Thank you for your comment. Primary prevention interventions were excluded because these are aimed at preventing violence before it occurs, therefore tends to be aimed at individuals or groups who have not necessarily experienced violence yet and therefore represents a different population than the one of interest for this review. We have added this justification into the eligibility criteria section.	Primary prevention services and interventions were excluded as they aim to prevent DSVA before it occurs, and therefore focus on individuals or populations who have not necessarily experienced violence.
11. Methods - Authors should discuss how general trauma intervention studies were handled – were they included/excluded?	The primary aim of the scoping review was to understand outcomes that were used in interventions and services for people who had experienced DSVA. For interventions/services with mixed populations, we used a cut of off 50%, where more than 50% of participants had to have experienced DSVA for the study to be included. This means that in theory, a general trauma intervention would have been included, if the majority of participants had experienced DSVA. This has been explained in the eligibility criteria section.	Interventions and services that were not specifically aimed at DSVA, such as general trauma interventions, were only included if the majority of participants (>50%) had experienced DSVA.
12. Methods - What search terms were used and how were they identified?	We followed PRISMA-ScR guidance (item 8) (Tricco, AC, Lillie, E, Zarin, W, O'Brien, KK, Colquhoun, H, Levac, D, Moher, D, Peters, MD, Horsley, T, Weeks, L, Hempel, S et al. PRISMA extension for scoping reviews	Appendix 2 Search terms were identified during initial scoping and protocol development stages. An initial list of search terms was drafted based on terms found in the literature and

	(PRISMA-ScR): checklist and explanation. Ann Intern Med. 2018;169(7):467-473. doi:10.7326/M18-0850.) and presented the full electronic for one of the peer reviewed and one of the grey literature search strategies in Appendix 2. These contain all the search terms that were used in the review. We have now updated this include an explanation of how we identified search terms, also in Appendix 2, Search strategy development process.	based on search terms used in systematic reviews in related areas. This initial list of terms was shared with the review team and with the Patient and Public Involvement advisory group, who added any additional terms thought to be relevant to the review's research question.
13. Methods - When data was missing, how long did article authors get to submit their responses?	We contacted authors again after two weeks if we had not heard back from them after we initially contacted them, and were then given a minimum of an additional two weeks. This has been added to the 'data charting process and data items' section.	After two weeks, if there had been no response, corresponding authors were contacted a second time and given a minimum of an additional two weeks to respond.
14. Results - Authors should elaborate on the type of health outcomes measured	All outcomes in evaluations and studies that were used to examine effectiveness were included in this scoping review, including any health outcomes, therefore there were no restrictions on the type of health outcomes included. Appendix 5 lists in detail the specific outcome measures that were identified, with outcomes in the 'health' category detailed on pages 1-5. We have added a sentence to the 'data charting process and data items' section to highlight that all and any outcomes were extracted.	All outcomes that were used to assess the effectiveness of DSVAs support interventions and services were extracted

15. Results - This sentence needs clarification: “Similarly, whilst the overarching category of DSVAs contained two primary domains: the experience of DSVAs and the perpetration of DSVAs, many outcome measures could fall into either domain, depending on who the respondent was (i.e., the perpetrator or the victim-survivor).”	Thank you. We have edited the sentence and added in an example for clarity. We hope this edit makes the sentence easier to understand.	Similarly, the overarching category of DSVAs contained two domains: the experience of DSVAs and the perpetration of DSVAs. However, many outcome measures could fall into either domain, depending on who the respondent was (i.e., the perpetrator or the victim-survivor). For example, the outcome ‘presence of abuse’ would fall into the ‘experience of DSVAs’ domain if the respondent was the victim-survivor, and the ‘perpetration of DSVAs’ domain if the respondent was the perpetrator.
16. Results - Page 15 has an inserted formatting error: (Appendix 5Error! Reference source not found.)	Thank you for highlighting this error, this has been corrected.	
17. Figure 1 needs to be reviewed and adjusted as the numbers don’t flow/add up in the current form. For example, why are the records screened (11571) different from total identified (17911)?	Figure 1 has been checked and we corrected two errors in the ‘identification of studies via other methods’ side of the diagram. The difference in numbers between the records identified and records screened is because of the number of records removed before screening as part of the deduplication process (n=6340).	
18. Discussion: Overall well developed. I would add to the limitations that only 20% of records were dually screened at each stage.	This has been added into the limitation section.	Additionally, because of time and resource restrictions, only 20% of records were dual screened during the title/abstract and full text screening stages, which does increase the risk of relevant studies being incorrectly excluded.

VERSION 2 – REVIEW

REVIEWER	Benjamin Hine University of West London
REVIEW RETURNED	28-Nov-2023

GENERAL COMMENTS	This manuscript presents a scoping review examining the outcomes used in evaluations of domestic, sexual, and gender-based violence (DSGBV) support interventions and services. It aims to understand which outcomes are being measured, how they are being measured, and whether these differ by type of support, DSGBV type, source of studies, and setting. The review included studies from January 2000 to March 2021, employing a comprehensive search strategy and following the PRISMA-ScR guidelines. The findings reveal a variety of outcomes measured across studies, highlighting the need for standardized outcome measures in DSGBV research. Introduction Strengths: Provides a thorough background on DSGBV and the importance of evaluating support interventions. Clearly outlines the research gap in outcome measurement. Suggestions: Consider expanding on the theoretical or conceptual framework guiding the review. Introduce earlier discussions on the diversity of outcomes in existing literature. Method Strengths: Methodology is detailed and well-articulated, following PRISMA-ScR guidelines. Comprehensive search strategy and clear inclusion/exclusion criteria. Suggestions: Discuss potential biases or limitations introduced by the chosen databases and search terms. Consider the impact of excluding non-English studies on the review's comprehensiveness. Results Strengths: Results are presented systematically, with a clear summary of the types of outcomes measured. Effective categorization of studies according to various criteria. Suggestions: Include more detailed analysis or discussion of the reasons behind the diversity of outcomes. Visual aids (charts, graphs) summarizing the study categorizations could enhance clarity. Discussion Strengths: Thoughtful integration of results with broader DSGBV research. Discussion on the implications for standardizing outcome measures in DSGBV research is insightful. Suggestions: Expand on how these findings could inform future research agendas. Offer more detailed suggestions for policymakers and practitioners
--

	in DSGBV fields. Overall Comments This manuscript makes a significant contribution to the understanding of outcome measurement in DSGBV research. The methodology is robust, and the presentation of results is clear and well-organized. The discussion provides meaningful insights into the implications of these findings. Enhancing the manuscript with more detailed theoretical insights, a deeper analysis of the diversity of outcomes, and more explicit suggestions for future research could further strengthen its contribution to the field.
--	--

REVIEWER	Ameeta Kalokhe Emory University
REVIEW RETURNED	30-Oct-2023

GENERAL COMMENTS	The authors have effectively addressed my questions/concerns with the first draft. I think this is a very well-designed, systematic, thorough scoping review, with key strengths being the inclusion of gray literature and and use of the advisory board. I have no further comments.
--

VERSION 2 – AUTHOR RESPONSE

Comment	Response	Changed text
Reviewer 2		
1. The authors have effectively addressed my questions/concerns with the first draft. I think this is a very well-designed, systematic, thorough scoping review, with key strengths being the inclusion of gray literature and and use of the advisory board. I have no further comments.	Thank you for your comment. We are pleased that we addressed your previous concerns and questions, and thank you for your feedback.	N/A
Reviewer 1		
2. This manuscript presents a scoping review	Thank you for your comment.	N/A

examining the outcomes used in evaluations of domestic, sexual, and gender-based violence (DSGBV) support interventions and services. It aims to understand which outcomes are being measured, how they are being measured, and whether these differ by type of support, DSGBV type, source of studies, and setting. The review included studies from January 2000 to March 2021, employing a comprehensive search strategy and following the PRISMA-ScR guidelines. The findings reveal a variety of outcomes measured across studies, highlighting the need for standardized outcome measures in DSGBV research.		
3. Introduction Strengths:	Thank you for your comment.	N/A

Provides a thorough background on DSGBV and the importance of evaluating support interventions. Clearly outlines the research gap in outcome measurement.		
4. Introduction Suggestions: Consider expanding on the theoretical or conceptual framework guiding the review.	We have expanded upon some of the theoretical and conceptual frameworks underlying DSGBV support interventions in the Introduction. There are several theories and frameworks that explain how these interventions might work to lead to particular outcomes; we have focused on the socio-ecological model of violence against women (Heise, 1998) and Sullivan's (2017) conceptual model for DV services, which draws upon Conservation of Resources theory.	Several theoretical frameworks have been developed to explain how such support services and interventions may work. For instance, the socio-ecological model of violence against women highlights key risk factors at the individual, relationship, community and societal levels, identifying potential intervention points for preventing and responding to violence [38]. One of the risk factors identified by Heise is social isolation, therefore one component of a support intervention may be to increase social support and build

		relationships. Sullivan's conceptual model of DV support services [39] builds on the Conservation of Resources theory, which suggests that psychological distress following trauma is influenced by the loss of economic, social and interpersonal resources central to their well-being [40-42]. Based on this, Sullivan provides an exemplar model for support services which outlines eight common programme activities aimed at creating communities that value their members and promote well-being, though intrapersonal and interpersonal changes, which impact a range of intrapersonal and interpersonal social and emotional well-being outcomes. Page 6
5. Introduction Suggestions:	Thank you for the suggestion. We have added a few sentences outlining some existing literature that focuses on	Previous research in the

Introduce earlier discussions on the diversity of outcomes in existing literature.	outcome measurement in this field, including a review of outcomes used in a specific sub-set of services/interventions, and several pieces that have discussed the various challenges of outcome measurement and defining success in this context.	UK and US literature that has explored outcome measurement in the domestic violence field has noted the diversity of reported outcomes (e.g. [52, 53]) and highlighted various issues and difficulties surrounding outcome measurement, which contributes this diversity [54, 55]. These studies point to the differing priorities of funders and service providers, and the diversity of specific service goals and objectives, which are often multiple and complex, as key drivers behind the range of outcomes being utilised. This literature also includes discussions on what should be measured, with several pushing for the extension and diversification of outcomes and measurement strategies, potentially
---	---	--

		contributing to the diversity of outcomes being used. Page 7
6. Method Strengths: Methodology is detailed and well-articulated, following PRISMA-ScR guidelines. Comprehensive search strategy and clear inclusion/exclusion criteria.	Thank you for your comment.	N/A
7. Method Suggestions: Discuss potential biases or limitations introduced by the chosen databases and search terms.	The search strategy was designed in a way to minimise bias as far as possible. For instance, search terms were checked with the stakeholder group and were as comprehensive as feasible. However, it is possible that some relevant terms were not included, potentially missing relevant papers. We also included both peer-reviewed and grey literature databases to ensure a wide spread of evidence was searched and to further minimise potential bias. The peer-reviewed literature databases covered a wide range of fields, including medicine, health/health care, behavioural sciences, sociology, health policy, psychology, social work, culture and ethnicity, politics, education, demography, economics, human services, social welfare, social policy, community development and social sciences and many more. Additionally, the grey literature search searched for reports from third sector/charity organisations, government reports and various other non-peer reviewed sources of evidence, and the call for evidence allowed organisations who may not have capacity to publish online to share with us any relevant reports or documents. However, we have added a sentence to the 'limitations' section of the discussion to acknowledge that some potentially relevant papers may have been missed due to the search strategy. In this section we have also noted a type of publication bias where reports and evaluations conducted by smaller organisations with fewer resources to publish are less likely to be identified. Finally, as a result of the next suggestion, we have added some text in the methods section	Further, whilst the search strategy was comprehensive and designed in collaboration with the advisory group to ensure key search terms were included, the search was not exhaustive, and therefore it is possible that had we selected different databases or included additional terms, we may have identified additional relevant papers. Page 26

	regarding the focus on English language and UK geography, which is also a limitation to the search strategy in terms of generalisability.	
8. Method Suggestions: Consider the impact of excluding non-English studies on the review's comprehensiveness.	While we note that the exclusion of non-English studies may impact the comprehensiveness of the review, given the focus of the review on UK-based support services, and that our inclusion/exclusion criteria was limited to papers reporting on services based in the UK, it was felt that the impact of limiting to English language studies would be minimal as we would expect relevant studies to be reported in English. However, we note that the focus on UK services and interventions does limit the generalisability of the results to outside of the UK context. We have added a brief sentence to explain this.	Only English language studies were eligible for inclusion. Given the focus on UK-based support services and interventions, we considered the impact of this restriction on the review's comprehensiveness to be minimal, however this does mean that the results of this review cannot be generalised to contexts outside of the UK. Page 9
9. Results Strengths: Results are presented systematically, with a clear summary of the types of outcomes measured. Effective categorization of studies according to various criteria.	Thank you for your comment.	N/A
10. Results Suggestions: Include more detailed analysis or discussion of the reasons behind the diversity of outcomes.	Thank you for the suggestion. We have extended the discussion of this, exploring potential reasons for this diversity, within the discussion section of the review.	Overall, this review has demonstrated that the types of outcomes measured when assessing the effectiveness of support

		interventions and services for people who have experienced DSVAs vary widely. There are several potential reasons for this diversity in outcome measures. Firstly, as previously mentioned, DSVAs have many impacts, including on physical health, mental health, housing, finances, relationships, and many more. Therefore, there are many avenues through which support services and interventions could have impact and improve outcomes. Indeed, in their review Sprague, McKay [52] noted that many services have multiple goals that cannot be assessed by a single outcome, thus multiple outcomes are required. Secondly, differences between the priorities of funders, services and service
--	--	---

		users, and researchers in terms of outcome measurement likely results in diversity in outcomes and differences in outcome measures utilised between sectors [53, 54]. Thirdly, there is currently a lack of consensus across the field regarding what outcomes should be measured and, importantly, how. For instance, while it may be agreed that 'safety' is an important outcome to assess, how this is operationalised and measured is not consistent. Page 25
11. Results Suggestions: Visual aids (charts, graphs) summarizing the study categorizations could enhance clarity.	While we have included a number of tables to summarise the study and outcome categorisations (Tables 2 and 3, and the tables in Appendix 4 and 5), we appreciate that these are not particularly visual. We currently include two figures to visually represent the categorisation of outcomes (previously Figure 2, now Figure 3) and the temporal trends in the outcomes (Previously Figure 3, now Figure 4- Appendix 6), and we have now added an additional figure (Figure 2) to visually represent the study categorisations.	Figure 2.
12. Discussion Strengths: Thoughtful integration of results with broader DSGBV research. Discussion	Thank you for your comment.	N/A

on the implications for standardizing outcome measures in DSGBV research is insightful.		
13. Discussion Suggestions: Expand on how these findings could inform future research agendas.	We currently include a paragraph on future research focused on how the findings from this scoping review could inform future systematic reviews and evidence syntheses, to strengthen the current evidence base on the impact of specialist support services for those experiencing DSVA. We have expanded this and made this more explicit on page 27.	The findings from this scoping review on outcomes most commonly measured to assess the effectiveness of DSVA interventions highlight how support services are working to promote survivors' safety and wellbeing. However, the diversity of the specific outcomes and measurement tools used has implications for researchers, service providers, policymakers and funders. Increased consensus between researchers, service providers, policymakers and funders is needed. This will allow for more meaningful syntheses of the literature, as well as building a

		larger evidence base, so that a better understanding of the most effective means of support for people who have experienced DSVAs can be reached. Thus, further research is needed to extend ongoing work on the development of shared outcome frameworks, to determine which outcomes and outcome measures are most appropriate, valid and relevant, in order to work towards a consensus and build a shared evidence base to enable future meta-analyses. These outcome frameworks should be underpinned by programme theories. We hope our findings will inspire further conversation and exploration of what to measure, when and with whom, and provide guidance to researchers, service providers and funders striving to strike
--	--	---

		the balance between reaching a shared consensus, and selecting outcomes according to important contextual factors such as the (often multiple and complex) goals of the intervention, the timing of evaluation, and the study population, for example. Page 27.
14. Discussion Suggestions: Offer more detailed suggestions for policymakers and practitioners in DSGBV fields.	Scoping reviews are conducted to identify knowledge gaps, scope a body of literature, clarify concepts, investigate research conduct, or to inform a systematic review (Munn 2018 https://bmcmmedresmethodol.biomedcentral.com/articles/10.1186/s12874-018-0611-x). We would be reluctant to provide direct suggestions for policy and practice on the basis of findings from a scoping review, but would suggest that our findings are used to inform the development of conceptual frameworks and the design of subsequent research such as systematic reviews (see point 13 above), which would in turn lead to tangible suggestions for policy and practice. Therefore, we have added a few broad suggestions for practitioners, while also acknowledging that they are overburdened and under-resourced and therefore want to avoid making multiple recommendations that are not feasible or useful in practice and cannot realistically be implemented. We have also added a few suggestions for policymakers/funders and commissioners, which were informed by discussions with our advisory group throughout the review process.	Implications and recommendations for service providers Whilst we recognise that service providers working in the DSVA field are often overburdened and under-resourced, and doing the best they can with what they have, we would encourage service providers to use the findings from this review to consider their current practices for outcome measurement and the how's and why's

		underlying them. If resource allows, organisations could assess whether the outcome measurement tools currently being used are fit for purpose and reflect recent shifts in definitions of and responses to domestic abuse, and explore opportunities for improvement and innovation. In the absence of additional resource, service providers should continue to record and monitor the various service- and individual-level outcomes they are required to measure as accurately and consistently as is possible whilst delivering frontline services. Implications and recommendations for policymakers Ultimately, the extent to which service providers can apply research findings to practice and use
--	--	--

		them to drive improvements to services depends heavily upon policymakers' decisions relating to the provision and funding of services, and monitoring and evaluation. Currently, patchwork and piecemeal funding [36, 196] and inconsistent commissioning frameworks contribute to the diversity of outcomes measured and underpin a competitive funding landscape that is not conducive to achieving consensus across the sector. Service providers working to end DSVAs have been campaigning for greater consistency in the commissioning of services for many years, and their voices should be used by policymakers to enact meaningful change. Based on discussions with our advisory
--	--	--

		group throughout the review process, we would suggest that policymakers focus on ensuring any core outcomes framework is implementable on a practical level. Service providers in low resource settings likely have less scope for this than others, and it is crucial that any core outcome measurement tools are not too time-consuming for service providers to use and complete. Additionally, policymakers and commissioners should look beyond blunt, immediately measurable outcomes (e.g. immediate safety, prosecution and conviction rates) and recognise the value of other outcomes (e.g. social and emotional wellbeing), which may take longer to measure but be a more accurate indication of real change. A
--	--	--

		suggested avenue for working towards this would be to run a national survivor consultation to draw out what is the most meaningful and/or has made the most difference for survivors. Key findings from this could then be mapped onto the outcomes prioritised by policymakers, service providers and funders, to identify where there are overlaps and where there are disparities that need addressing. Pages 28-29.
15. Overall Comments This manuscript makes a significant contribution to the understanding of outcome measurement in DSGBV research. The methodology is robust, and the presentation of results is clear and well-organized. The discussion	Thank you for your comment, and we hope that the revisions outlined above have adequately addressed your suggestions.	N/A

provides meaningful insights into the implications of these findings. Enhancing the manuscript with more detailed theoretical insights, a deeper analysis of the diversity of outcomes, and more explicit suggestions for future research could further strengthen its contribution to the field.		
--	--	--

VERSION 3 – REVIEW

REVIEWER	Benjamin Hine University of West London
REVIEW RETURNED	06-Feb-2024

GENERAL COMMENTS	General Suggestions: Clarity and Consistency: Ensure consistency in the terminology used for DSVAs interventions and outcome measures across the manuscript. Methodological Limitations: Discuss any limitations related to the scoping review methodology, such as potential biases in selecting and interpreting studies. Future Research Directions: Suggest specific areas for future research, such as the development and validation of standardized outcome measures for DSVAs interventions. Practical Implications: Expand on how the findings can inform policy and practice, including recommendations for stakeholders involved in DSVAs service provision.
--